# The Valence and Spin State Tuning of Iron(II/III) Porphyrazines with Bulky Pyrrolyl Periphery in Solution and Solid State

**DOI:** 10.3390/molecules27227820

**Published:** 2022-11-13

**Authors:** Tomasz Koczorowski, Wojciech Szczolko, Pawel Bakun, Barbara Wicher, Lukasz Sobotta, Maria Gdaniec, Anna Teubert, Jadwiga Mielcarek, Ewa Tykarska, Jozef Korecki, Kvetoslava Burda, Tomasz Goslinski

**Affiliations:** 1Chair and Department of Chemical Technology of Drugs, Poznan University of Medical Sciences, Grunwaldzka 6, 60-780 Poznan, Poland; 2Chair and Department of Inorganic and Analytical Chemistry, Poznan University of Medical Sciences, Rokietnicka 3, 60-806 Poznan, Poland; 3Faculty of Chemistry, Adam Mickiewicz University, Uniwersytetu Poznańskiego 8, 61-614 Poznan, Poland; 4Institute of Bioorganic Chemistry, Polish Academy of Sciences, Z. Noskowskiego 12, 61-704 Poznan, Poland; 5Jerzy Haber Institute of Catalysis and Surface Chemistry, Polish Academy of Sciences, Niezapominajek 8, 30-239 Krakow, Poland; 6Faculty of Physics and Applied Computer Science, AGH University of Science and Technology in Krakow, al. Mickiewicza 30, 30-059 Krakow, Poland

**Keywords:** porphyrazine, crystal structure, Mössbauer spectroscopy, iron valence and spin state

## Abstract

Iron(III) porphyrazines containing peripheral 2,5-dimethyl-, 2-methyl-5-phenyl-, and 2,3,5-triphenyl-1*H*-pyrrol-1-yl substituents were synthesized and subjected to physicochemical characterization. This was accomplished by high-resolution mass spectrometry, nuclear magnetic resonance (as diamagnetic Fe(II) derivatives), HPLC purity analysis, and UV-Vis spectroscopy, accompanied by the solvation study in dichloromethane and pyridine. X-ray structure analysis was performed for a single crystal of the previously obtained 2,5-diphenyl-substituted derivative of porphyrazine complex (**5d**). The octahedral geometries of iron cation, present in the porphyrazine core, influenced the packing mode of molecules in the crystals. Mössbauer studies, performed for solid samples of iron porphyrazines, indicated that low-spin reduced iron states might occupy low- or high-symmetry binding sites. It was found that the hyperfine parameters and the subsequent contribution of the iron cations depend on the number of phenyl groups surrounding the pyrrolyl moiety. For iron(II) porphyrazine 2,3,5-triphenylpyrrol-1-yl substituents (**5b**), a high-spin ferrous state fraction was observed. Temperature-dependent measurements showed that the freed rotation of the peripheral porphyrazine ligands and the increased flexibility of the macrocycle ring result in the Fe^2+^ ion being stabilized in a diamagnetic state at a binding site of high symmetry at room temperature in the solid state. This process is most probably stimulated by the range of collective motions of the polymeric ribbons consisting of iron(II) porphyrazines observed in the X-ray.

## 1. Introduction

Porphyrazines (Pzs) along with their structurally relevant analogs, such as porphyrins and phthalocyanines, belong to porphyrinoid macrocycles. Due to their unique photo and electrochemical properties, many of them revealed potential applications as photosensitizers in medicine and as catalysts or sensors in technology [1,2]. Pzs can be modified by exchanging the metal cation inside the core or by peripheral modifications with aliphatic, aryl, sulfur, oxygen, and nitrogen groups in their β positions [3,4,5,6,7]. Some Pzs metalized with iron Fe^2+/3+^ cations have demonstrated catalytic properties in oxidation reactions [8,9,10]. Our group reviewed electrochemical and catalytic activity as well as sensing properties of various iron(II/III) porphyrazines and phthalocyanines [11,12]. The influences of peripheral and axial substituents of iron(II/III) porphyrins and porphyrazines on their UV-Vis spectra were reviewed and investigated by Nakamura and Stuzhin [13,14,15]. The aforementioned phenomenon impacts further applications of iron(II/III) porphyrazines. Therefore, it is crucial to determine the Fe spin state and tune it by axial ligation.

There have been limited data on Pzs modified with peripheral 5-membered heteroaromatic rings. Luo et al. [16] synthesized binuclear Pzs with twelve peripheral trimethylthiophenyl groups, revealing interesting photochromic characteristics. For the last 15 years, our group obtained many examples of Pzs bearing peripheral 2,5-dimethylpyrrol-1-yl, 2,5-dithienylpyrrol-1-yl, 2,5-di(biphenyl-4-yl)pyrrol-1-yl, 2-(1-adamantyl)-5-phenylpyrrol-1-yl, and 3,4-dihalide-2,5-dimethylpyrrol-1-yl moieties [17,18,19,20,21]. Lately, it was found that bulky 2,5-diphenylpyrrol-1-yl substituents on the periphery of iron(II) porphyrazine influence its physicochemical properties, which was confirmed by Mössbauer spectroscopy [22]. The current study presents a series of porphyrazines with pyrrolyl groups substituted in their second, third, and fifth positions with methyl and/or phenyl groups. Novel compounds were characterized in terms of their structural properties using X-ray crystallography, NMR, and Mössbauer spectroscopy. The valence and spin states of the iron cation inside the macrocyclic core in solution and solid state were discussed.

## 2. Results and Discussion

### 2.1. Synthesis and Physicochemical Characterization

Previously, we reported the synthetic pathway and physicochemical characterization of metal-free porphyrazines with an alternate system of 2,3,5-trisubstituted pyrrolyl and dimethylamino substituents [18,22,23]. The Paal–Knorr reaction of diaminomaleonitrile (**1**) with 1-phenylpentane-1,4-dione in methanol, 1,2,4-triphenylbutane-1,4-dione in methanol, and 2,5-hexanedione in benzene, and in the presence of the catalytic amount of trifluoroacetic acid (TFA) or oxalic acid, led to the maleonitrile derivatives **2a**, **2b**, and **2c**, respectively (Figure 1). This synthetic approach leading to pyrrolyl-substituted maleonitriles was discovered by Begland et al. and further developed by our group [24]. Compounds **2a**–**2c** were subjected to alkylation reactions following the procedure used by Beall et al. [25] to give **3a**–**3c**. Next, the maleonitrile derivatives were utilized in the macrocyclization reaction in dimethylaminoethanol (DMAE) following the procedure used by Decreau et al. [26], leading to the free base porphyrazines **4a**, **4b,** and **4c**. Their subsequent metalation with FeBr_2_ in the toluene-THF-2,6-lutidine system [27] allowed the obtaining of corresponding novel iron Pzs **5a**, **5b**, and **5c** with 54%, 58%, and 46% yield, respectively.

Synthesis of **5a**, **5b**, and **5c** may be considered as the continuation and further development of the recently published methodology which allowed the obtaining of magnesium(II) porphyrazine with 2,5-diphenylpyrollyl and dimethylamino groups and its iron(II) analog **5d** [18,22].

All new macrocycles **5a**–**5c** were purified by column chromatography and subjected to physicochemical characterization, including high-resolution mass spectrometry, nuclear magnetic resonance, HPLC purity analysis, and UV-Vis spectroscopy, accompanied by a solvation study in dichloromethane and pyridine.

In the UV-Vis spectra of **5a**–**5c** recorded in dichloromethane, the Soret and Q-bands characteristic for porphyrinoids can be observed at approximately 300 nm and 690 nm, respectively (Figure 1). The obtained data were in good agreement with those obtained for the 2,5-diphenylpyrrolyl derivative reported earlier, where the Soret and Q-band were detected at 296 nm and 691 nm, respectively [22]. As a consequence of using the preparation method involving purification of synthesized macrocycles from iron salt excess in a two-phase system of dichloromethane:1M HCl, all the obtained UV-Vis spectra are similar to those recorded for five-coordinated Fe(III)Cl porphyrazine species [28,29] with an additional red-shifted band at approximately 915 nm. It can be noted that the addition of phenyl groups to peripheral pyrrolyl substituents in **5a** and **5b** led to an increase in the Soret/Q-band ratio compared to **5c** with 2,5-dimethylpyrrolyl moieties.

The solvation studies of **5a**–**5c** performed in dichloromethane and pyridine revealed the strong axial coordination of pyridine molecules to the central metal cation (Appendix A). The decline of additional red-shifted peaks in UV-Vis spectra was observed, indicating the change in iron cation valence from paramagnetic five-coordinated Fe(III) complex to diamagnetic six-coordinated Fe(II) complex. It is known that in solvents with higher dielectric constant and/or donor properties (acetone, dimethylsulfoxide, pyridine, etc.), Fe^3+^ is reduced to Fe^2+^ with the formation of diamagnetic six-coordinated iron(II) complexes (L)_2_FePz [14] which can be noticed by changes in the UV-Vis spectra as in this case. In addition, the synthesized porphyrazines obeyed the Lambert-Beer law in pyridine, whereas aggregation could be observed in freshly distilled dichloromethane (Appendix A). Moreover, we performed an additional experiment with the formation of the µ-oxo dimer of Fe(III)Pz, following the literature procedure [30]. The porphyrazine **5c** was subjected to an oxidation reaction in a two-phase system: dichloromethane:1M NaOH, exposed to the air while stirring. In the UV-Vis spectra collected in dichloromethane, two effects indicating the formation of the dimeric form were noted [14]. The first was a decay of the peak at 916 nm, and the second was a red-shifting of the Q-band from 685 nm to 782 nm (Figure 2).

The ^1^H and ^13^C NMR spectra of synthesized porphyrazines **5a**–**5c** were recorded in deuterated pyridine for better solvation than in CDCl_3_, resulting in a reduction of the paramagnetic Fe(III) species to the diamagnetic Fe(II) derivative. In the ^1^H NMR of **5a**, the splitting of CH_3_ signals from both dimethylamino and 2-methyl-5-phenylpyrrolyl groups can be noticed in the aliphatic area due to atropoisomer formation (see page 18 in Appendix A) observed for the magnesium(II) analog [18]. The occurrence of the ^1^H signal splitting of phenyl substituents in the aromatic area in **5b** due to atropoisomerism was also recorded. However, the ^1^H NMR spectrum of **5c** revealed the existence of two forms of porphyrazine macrocycle—a free, non-aggregated Pz **5c** and an aggregated Pz **5c*** (see page 20 in Appendix A). Similar behavior was reported earlier for porphyrazine **5d** with peripheral dimethylamino and 2,5-diphenylpyrrolyl substituents [22]. The assignment of ^1^H and ^13^C NMR signals and the nuclear correlations of porphyrazines **5a**–**5c** are presented in Appendix A.

### 2.2. X-ray Study

All iron(III) porphyrazines **5a**–**5c** obtained in this work and the fourth analog of the series, previously synthesized Pz **5d,** were subjected to crystallization experiments, resulting in the formation of a fine single crystal for the latter. The crystal of Pz **5d** (Figure 3) was obtained at room temperature by slow evaporation of dichloromethane:methanol (1:1) resulting in the substitution of chloride with methanol moieties in the porphyrazine complex. Crystal data and details of the X-ray diffraction analyses are reported in Appendix A.

X-ray structural analysis revealed that in the crystal of **5d**, the Fe(II)-Pz molecule is located at a special position of S_4_ symmetry. The iron cation is in a slightly distorted octahedral environment with four N-pyrrole atoms located in equatorial positions and two O atoms from the coordinated methanol ligands in apex positions, indicating the presence of a six-coordinated Fe^2+^ cation. The Fe(II)-N and Fe(II)-O bond lengths are 1.926(2) and 2.208(3) Å, respectively. The Pz macrocycle exhibits a saddle-like conformation with the displacement of the β-pyrrole carbons from the Pz best plane in the range ±0.371(3) Å (Figure 4 and Appendix A).

Large substituents attached to the Cβ atoms of the pyrrole rings lead to steric overcrowding of the periphery of the Pz molecule. Hence, the peripheral groups are twisted with respect to the mean plane of the Pz core, and the rotations about the Cβ-N bonds differ depending on the substituent [18]. The dihedral angles formed with the Pz core by the dimethylamine groups [34.9(1)°] are smaller than those created by the exo-pyrrole rings [84.1(3)°]. A Cβ-N bond length to the dimethylamine group of 1.361(3) Å points to a significant electron conjugation between the Pz core and this group. This conjugation is strongly reduced for the exo-pyrrole ring, as reflected in the Cβ-N bond length of 1.420(4) Å.

In **5d**, hydrophobic pockets are generated on both sides of the macrocycle ring by substituents strongly twisted relative to the Pz core. In the case of octahedral coordination, both pockets are occupied by a coordinated solvent molecule. In magnesium(II) porphyrazines with similar pyrrolyl substituents, the central atom is in the square-pyramidal coordination. Only one of the pockets is occupied by a coordinated solvent molecule. Thus, porphyrazine molecules associate with centrosymmetric dimers through the interpenetration of the phenyl rings into unoccupied pockets [18]. In **5d,** such dimers cannot be formed as both pockets are filled with solvent molecules. Instead, a ribbon motif formed by porphyrazine molecules is observed (Figure 5).

### 2.3. Mössbauer Spectroscopy Study

The solid samples of synthesized porphyrazines **5a**–**5c** containing peripheral 2,5-disubstituted- and 2,3,5-trisubstituted-pyrrol-1-yl groups, obtained by slow evaporation of dichloromethane:methanol (1:1, *v*/*v*) mixture, were researched by Mössbauer spectroscopy. As mentioned in Section 2.2, the use of methanol can cause the substitution of chloride anions with MeOH moieties. Mössbauer spectroscopy is a unique method that allows directly detecting local valence and spin states of heme-iron in Pzs. The hyperfine parameters, an isomer shift (IS), and a quadrupole splitting (QS) obtained from theoretical fits of the experimental data are additionally sensitive to the type of iron porphyrinoids and their arrangement [31,32]. The preparation of solid samples involving the solvent with a high dielectric constant (methanol) has a strong influence on the structure of porphyrazine complexes due to obtaining low-spin six-coordinated Fe^2+^ species [14,33] or intermediate-spin Fe^3+^ species [13].

In Figure 6, the Mössbauer spectra of **5c** measured at 95 and 304 K are presented. Three doublets having small comparable isomer shifts (~0.17 mm/s at low temperatures) but different quadrupole splittings indicate three distinct configurations of Fe binding sites. In all cases, heme-iron is stabilized in a low-spin ferrous state (in a diamagnetic state, Fe^2+^ with S = 0). Electrons from the nitrogen atoms of dimethylamino groups donate electrons to the iron ion, keeping it in a reduced Fe^2+^ state. Usually, one could expect a small QS (about 0.1–0.4 mm/s) for such a diamagnetic iron state as is observed for component #3 (comp#3) (Appendix A) [32,34]. However, distortion of the octahedral field in the iron-binding site significantly increases quadrupole splitting. Even identical ligands having varying distances from the central ion lead to a reduction in the field symmetry. A different character of one of the axial ligands can also cause the deformation of the octahedron. Heme-iron in oxyhemoglobin is such an example [32,35]. Additionally, the growth of the quadrupole splitting can be enhanced by a torsion of the tetrapyrrole ring [22,36]. Therefore, the two other components detected for **5c**, comp#1 and comp#2, characterized by a QS as high as ~2.8–3.3 mm/s, may also be assigned to the low-spin Fe^2+^ state, especially as the temperature dependence of their hyperfine parameters and contributions confirm our hypothesis. A substantial decrease in these quadrupole splittings with increasing temperature is related to the activated rotational flexibility of the iron axial ligands and the activation of short and long cooperative motions of the complexes [37,38]. Interestingly, the contribution of comp#1 starts to decrease at T > 160 K, whereas that of comp#2 begins to increase (Figure 6 and Figure 8, Appendix A). Additionally, at T > 185 K, the content of comp#3 increases. Because similar Debye temperatures are expected for all the Fe substates (~180–200 K) [31,37], one can conclude that comp#1 gradually transforms into comp#2 and then to comp#3, characterized by the highest symmetry of the electric field at the iron-binding site.

The Mössbauer spectra of the Pz with 2-methyl-5-phenylpyrrol-1-yl groups **5a** consist of three subspectra characterized by hyperfine parameters similar to those detected for **5c** (Figure 7 and Appendix A). However, the contributions of the subcomponents differ from those observed in the case of **5c**. Comp#2 has the highest content in the spectrum of **5a**. It increases from ~52% at 90 K up to ~61% at 200 K and decreases at T > 260 K but still dominates the other subcomponents. Within the range of temperatures from 90 K to 200 K, the contribution of comp#1 decreases from about 42% to 27%, whereas the contribution of comp#3 increases from about 6% up to about 12%. One can again notice that comp#1 transfers to comp#3 via comp#2. This effect is evident at T > 250 K, when the abrupt decrease in comp#2 content is related to the increase in comp#3 content (Appendix A and Figure 8).

From the data presented above, it can be concluded that the peripheral phenyl group is responsible for the inversion of the occupation of binding sites #1 and #2 by the iron atom. In addition, the interactions of the Pz ring and probably axial ligands with the phenyl groups force a more efficient formation of the high-symmetry field (comp#3) at temperatures above 200 K (Figure 8). The attachment of 2,5-diphenylpyrrol-1-yl groups to the porphyrazine core in **5d** results in an increase in the contribution of the most symmetric iron-binding site in the Mössbauer spectrum to about 36% at 85 K [22]. In the case of **5b**, in which 2,3,5-triphenylpyrrol-1-yl groups are present, the content of the highly symmetric iron-binding site (comp#2) reaches even 50% at 80K (Figure 9, Appendix A). The values of hyperfine parameters characterizing comp#1 for **5b** are weighted averages of those obtained for comp#1 and comp#2 in the case of the other Pzs **5a**, **5c,** and **5d**. Comp#3 observed for **5b** is characterized by a high-spin state. Its relatively small quadrupole splitting, smaller by about 0.6 mm/s than that detected for the high-spin state in **5d** [22], indicates higher field symmetry at the iron site in **5b**.

Mössbauer results show that the successive attachment of phenyl groups to the pyrrol-1-yl substituent enhances the stabilization of Fe^2+^ in a high-symmetry field. Its contribution linearly increases with an increasing number of phenyl groups bound to the pyrrole ring (Figure 10). A deviation from the rule at low temperatures is only observed for **5a** having only one phenyl group attached to the pyrrol-1-yl substituent and most probably comes from the presence of an extra methyl group, which can donate electrons to the central ion and modify the electron density distribution in the iron vicinity. Furthermore, the incorporation of 2-methyl-5-phenyl-pyrrol-1-yl groups at the Cβ positions of Pz pyrrole rings in comparison to more sterically bulky 2,5-diphenylpyrrol-1-yl and 2,3,5-triphenylpyrrol-1-yl groups decreases the barrier to their rotation about Cβ-N bonds. It has also been found that more than one phenyl group attached to the pyrrol-1-yl moiety influences the appearance of a high-spin state of ferrous iron at low temperatures, and its content in **5b** (2,3,5-triphenylpyrrol-1-yl group) is about 2-fold higher than that reported for **5d** (2,5-diphenylpyrrol-1-yl group) [22] (Figure 10).

Narrow line widths of all the subspectra resolved for the Fe-porphyrazines (about 0.18 ± 0.02 mm/s) suggest a high homogeneity of the heme-iron surroundings. It was proved that the temperature changes of the hyperfine parameters and fractions of the subspectra are reversible. This means that the iron ligands are conserved, and only their configuration is modified. High values of quadrupole splittings observed in **5c** at a low temperature for more than 90% of the iron fraction suggest that the electric field gradient at the six-coordinated iron (four bonds with N atoms from the tetrapyrrole ring and two axial bonds to methanol molecules) has tetragonal distortion and that it has high symmetry only for less than 10% of iron-binding sites. Interestingly, it was observed that with increasing temperature, more iron ions occupy a symmetric configuration. This is enhanced by the presence of a phenyl substituent at the pyrrol-1-yl group in **5a**. According to the Jahn-Teller theorem, there always exists a non-totally symmetric configuration compared to which the fully symmetric structure is unstable in transition metal complexes [39,40]. Even a minimal asymmetry of the field leads to the split of the degenerate ground state into two lower-lying nondegenerate states. Because the tetragonal distortion can lead to an octahedron either elongated or compressed, three possible energy minima are separated from one another by three potential barriers. If the potential barrier is ΔE > k_B_T, then a static effect is observed, and if ΔE ≤ k_B_T, a dynamic effect is observed. Thus, in the Mössbauer studies, three possible field configurations could be observed at sufficient or relatively low temperatures [40]. At higher temperatures, one could expect an increased occupation of the higher energetic state-which is the most symmetrical. The presented Mössbauer data show that diverse Fe species occur in all the Fe-Pzs investigated by us, and at room temperatures, the occupation of the Fe octahedral configuration increases.

On the basis of changes in the contribution of the component characterized by high QS in Mössbauer spectra as a function of temperature, taking into account the quadrupole splitting distribution (Appendix A), one can estimate the activation energy of the dynamic Jahn–Teller effect for Pz **5a** and Pz **5c**. Activation of fast collective motions at higher temperatures ultimately leads to the relaxation of the system. Therefore, the changes presented in Appendix A are well described by the simplified Debye model valid for *T* > *½ θ_D_* extended for anharmonicity of the lattice vibrations which leads to the temperature-dependent Debye temperature (*θ_D_* = *θ*_0_(1 + *AT*), where *θ_0_* is the Debye temperature at the low-temperature limit and *A* is a parameter of the effective Debye temperature variation) [37,41]. We obtained *θ_D_* = 213 ± 21 K (A = −(2.8 ± 0.3) × 10^−3^ K^−1^) and *θ_D_* = 134 ± 16 K (*A* = −(2.3 ± 0.4) × 10^−3^ K^−1^) for Pz **5c** and Pz **5a**, respectively, allowing the estimation of the activation energy 18 ± 2 meV for Pz **5c** and 12 ± 1 meV for Pz **5a**.

Our results also provide experimental evidence that the phenomenon described above is related to the dynamic Jahn–Teller effect [39,40], which can be additionally regulated by the arrangement of all ligands, including the phenyl groups attached to the pyrrol-1-yl substituents. Their presence may change electron density distribution in the iron surrounding and thus the metal-to-ligand distances as well as the bond angles. The electronic anisotropy can be reduced or enhanced by the intermolecular interactions between the phenyl-phenyl rings and/or phenyl-porphyrazine rings which occur in the case of J-aggregates. In the present study, it was shown that these effects might modify energetic barriers between the different configuration states of Pzs and in this way influence their stability at a given temperature. Most probably, the high-spin state detected for Pz **5b** at low temperatures also results from those interactions.

## 3. Materials and Methods

### 3.1. General Procedures

All reactions were conducted in oven-dried glassware under argon. All solvents were evaporated in a rotary evaporator at or below 50 °C. Reaction temperatures reported refer to external bath temperatures. Methanol, tetrahydrofuran, and dichloromethane were distilled before use. Other solvents and all reagents were obtained from commercial suppliers and used without further purification unless otherwise stated. Melting points (M_p_) were obtained on a “Stuart” Bibby apparatus and are uncorrected. The flash column chromatography was carried out on Merck silica gel 60, particle size 40–63 µm, and Fluka silica gel 90 C_18_-reversed phase. Thin layer chromatography (TLC) was performed on silica gel Merck Kieselgel 60 F_254_ plates and visualized with UV (λ_max_ 254 or 365 nm). UV-Vis spectra were recorded on a Hitachi UV/VIS U-1900 spectrometer (Hitachi High-Tech Corporation, Tokio, Japan); λ_max_ (log ε), nm. HPLC analyses were performed on an Agilent 1200 instrument (Agilent Technologies, Santa Clara, CA, USA)equipped with a DAD detector. The chromatographic separation was carried out on an Eclipse XDB-C_18_ column, 150 mm × 4.6 mm, 5 μm (Agilent) or Gemini C_6_-Phenyl column (Phenomenex), 150 mm × 4.6 mm, 5 μm using isocratic elution conditions at a flow rate of 1.0 mL/min in three different configurations of solvents. Mass spectra HRMS (MALDI TOF, Impact HD, Bruker Daltonics, Billerica, MA, USA)) were measured at the Wielkopolska Centre for Advanced Technologies in Poznan.

### 3.2. Synthesis

#### General Synthetic Procedure

The syntheses of porphyrazines **5a**–**5c** were performed following the procedures described earlier [18,22,23]: free-base porphyrazine derivative (**4a**: 48 mg, 0.043 mmol; **4b**: 25 mg, 0.015 mmol; **4c**: 84 mg, 0.09 mmol), FeBr_2_ (93 mg, 0.43 mmol for **5a**; 32 mg, 0.15 mmol for **5b**; and 194 mg, 0.9 mmol for **5c**), and 2,6-lutidine (2 mL) were heated in a toluene–THF mixture (1:1, 10 mL) under reflux for 20 h. After being allowed to cool to room temperature, the green mixture was evaporated to dryness. Next, a dark green solid dissolved in dichloromethane (10 mL) was mixed with 1M HCl (10 mL) for 30 min (two-phase system). The organic layer was mixed with brine (10 mL) for 30 min (two-phase system). The organic layer was separated, dried with anhydrous MgSO_4_, and evaporated in a rotary evaporator. The residual iron species were removed by extraction in CH_2_Cl_2_ (10 mL) and saturated citric acid solution (10 mL). The dry residue was extensively purified by column chromatography (CH_2_Cl_2_:CH_3_OH, 50:1) to give the targeted porphyrazines **5a**–**5c**.

*[2,7,12,17-Tetrakis(dimethylamino)-3,8,13,18-tetrakis-(2-methyl-5-phenyl-1H-pyrrol-1-yl)porphyrazinato]iron(III) chloride* (**5a**): dark green solid (27 mg, 54% yield). m.p. > 300 °C. R*_f_* (CH_2_Cl_2_:CH_3_OH, 50:1) 0.26. UV-Vis (CH_2_Cl_2_): λ_max_, nm (log ε) 289 (4.97), 377 (4.44), 434 (4.45), 553 (4.32), 689 (4.60), 917 (4.21). ^1^H NMR (500 MHz; pyridine-*d*_5_): δ_H_, ppm 2.10–2.78, 3.25–3.78 (2×m, 36H, pyrrolyl-CH_3_ and N(CH_3_)_2_), 6.57–6.93 (m, 20H, C_6_H_5_), 7.58–7.65 (m, 8H, C_6_H_5_). ^13^C NMR (125 MHz; pyridine-*d*_5_): δ_C_, ppm 13.1, 14.1, 14.3, 39.0, 39.9, 42.5, 108.2, 109.0, 109.1, 109.3, 110.3, 126.3, 127.0, 127.5, 127.7, 128.0, 128.8, 128.9, 129.6, 129.9, 135.7, 136.5, 138.1, 149.9. HRMS (MALDI TOF): *m*/*z* Calc. for C_68_H_64_N_16_Fe [M]^+^ 1160.4849, found 1160.6684. HPLC purity 97.97–98.57%.

*[2,7,12,17-Tetrakis(dimethylamino)-3,8,13,18-tetrakis-(2,3,5-triphenyl-1H-pyrrol-1-yl)porphyrazinato]iron(III) chloride* (**5b**): dark green solid (15 mg, 58% yield). m.p. > 300 °C. R*_f_* (CH_2_Cl_2_:CH_3_OH, 50:1) 0.33. UV-Vis (CH_2_Cl_2_): λ_max_, nm (log ε) 298 (5.24), 378 (4.59), 436 (4.58), 547 (4.41), 692 (4.69), 915 (4.37). ^1^H NMR (500 MHz; pyridine-*d*_5_): δ_H_, ppm 2.83–3.69 (m, 24H, N(CH_3_)_2_), 6.54–7.07 (m, 26H, C_6_H_5_), 7.15–7.16 (d, 1H, C_6_H_5_), 7.24–7.26 (m, 6H, C_6_H_5_), 7.30–7.40 (m, 15H, C_6_H_5_), 7.45–7.50 (m, 4H, pyrrolyl-H), 7.61 (s, 1H, C_6_H_5_), 7.67–7.71 (m, 9H, C_6_H_5_), 7.77–7.78 (m, 2H, C_6_H_5_). ^13^C NMR (125 MHz; pyridine-*d*_5_): δ_C_, ppm 39.5, 41.8, 110.0, 110.1, 110.2, 124.3, 125.9, 126.2, 126.3, 127.0, 127.1, 127.7, 127.8, 127.9, 128.0, 128.2, 128.3, 128.5, 129.0, 129.2, 131.1, 131.3, 131.4, 131.5, 132.4, 132.4, 133.8, 133.9, 134.3, 134.5, 134.6, 137.6, 137.7, 137.8, 145.5, 147.2, 149.4. HRMS (MALDI TOF): *m*/*z* Calc. for C_112_H_89_N_16_Fe[M + H]^+^ 1713.6806, found 1713.1149. HPLC purity 95.07–96.51%.

*[2,7,12,17-Tetrakis(dimethylamino)-3,8,13,18-tetrakis-(2,5-dimethyl-1H-pyrrol-1-yl)porphyrazinato]iron(III) chloride* (**5c**): dark green solid (41 mg, 46% yield). m.p. > 300 °C. R*_f_* (CH_2_Cl_2_:CH_3_OH, 50:1) 0.29. UV-Vis (CH_2_Cl_2_): λ_max_, nm (log ε) 310 (4.75), 373 (4.53), 429 (4.53), 552 (4.41), 685 (4.70), 916 (4.26). ^1^H NMR (500 MHz; pyridine-*d*_5_): δ_H_, ppm 1.99 (s, 24H, pyrrolyl-CH_3_), 2.20–2.33 * (m, pyrrolyl-CH_3_), 2.74 (s, 24H, N(CH_3_)_2_), 3.39–3.50 * (m, N(CH_3_)_2_), 5.85 (s, 8H, pyrrolyl-H), 6.27–6.34 * (m, pyrrolyl-H). ^13^C NMR (125 MHz; pyridine-*d*_5_): δ_C_, ppm 13.4, 14.4 *, 40.2 (N(CH_3_)_2_), 42.8 * (N(CH_3_)_2_), 107.0 *, 107.8, 132.3, 139.4, 151.8, 152.8, 165.2. An asterisk (*) indicates the aggregated specie. HRMS (MALDI TOF): *m*/*z* Calc. for C_48_H_56_N_16_Fe[M]^+^ 912.4223, found 912.6889. HPLC purity 94.58–100%.

### 3.3. Solvation Study

A solvation study was performed in dichloromethane and pyridine (Baker, J.T. Avantor, Allentown, PA, USA) according to a previously described method [42].

### 3.4. Single-Crystal X-ray Structure Determination

The X-ray diffraction measurements were carried out with an Oxford Diffraction Xcalibur E diffractometer with graphite-monochromated MoKα radiation (Oxford Diffraction Ltd, Abingdon, UK)). CrysAlis PRO [43] was used for data processing. Structures were solved by direct methods with SIR2004 [44] and refined by full-matrix least squares based on F^2^ (SHELXL-2016) [45]. All non-H atoms were refined with anisotropic displacement parameters. C-bound hydrogen atoms were placed at idealized positions and were refined as riding on their carriers with U_iso_(H) = 1.2U_eq_ for CH and CH_2_ groups and U_iso_(H) = 1.5U_eq_ for the methyl group.

Due to the S_4_ symmetry of the Pz molecule, the carbon atom of the coordinated methanol molecule is disordered and refined with an occupancy of 0.5. The second methanol molecule observed in the Pz pocket also had to be refined as disordered with an occupancy factor of 0.5. The positions of OH hydrogen atoms of methanol molecules were not determined. Other solvent molecules were severely disordered and could not be reliably located. The SQUEEZE [46] procedure was applied to remove the contribution of disordered solvents to structure factors using the following parameters: grid 0.20 Å, probe radius 1.2 Å, and NStep 6. Solvent-accessible voids in the unit cell have a volume of 1264.6 Å^3^, and the electron count per unit cell is 284.

Crystal data and details of data collection, data reduction, and the refinement process are given in Appendix A. Crystallographic data for this paper can be obtained free of charge from The Cambridge Crystallographic Data Centre via www.ccdc.cam.ac.uk/data_request/cif (CCDC 1570662) (accessed on 3 October 2022).

### 3.5. NMR Study

^1^H NMR and ^13^C NMR spectra were recorded using Bruker 400, 500, and 700 spectrometers. Chemical shifts (δ) are quoted in parts per million (ppm) and refer to a residual solvent peak. Coupling constants (*J*) are quoted in Hertz (Hz). The abbreviations *s*, *d*, *t*, and *m* refer to singlet, doublet, triplet, and multiplet, respectively. Chemical shifts of aggregated species are signed with an asterisk (*). Additional techniques (^1^H-^1^H COSY, ^1^H-^13^C HSQC, and HMBC) were used to assist allocation.

### 3.6. Mössbauer Spectroscopy Study

Porphyrazine derivatives containing iron ions were examined using Mössbauer spectroscopy. These measurements were carried out in a home-built cryostat. All samples were kept in darkness. ^57^Co(Rh), 50 mCi, was a source of 14.4 keV radiation. Porphyrazine **5a** and **5c** powder samples were investigated in a wide range of temperatures from 90 K to 304 K. The Mössbauer spectrum of powder **5b** was collected at 80 K. The temperature was stabilized within 0.03 K. Recoil software was used to evaluate experimental data [47]. Values of isomer shifts are given relative to α-Fe at 295 K.

## 4. Conclusions

Iron(III) porphyrazines with 2,5-dimethyl-, 2-methyl-5-phenyl-, and 2,3,5-triphenyl-1*H*-pyrrol-1-yl substituents at the periphery were synthesized. Physicochemical characterization was conducted with high-resolution mass spectrometry (MALDI TOF), nuclear magnetic resonance (^1^H, ^13^C, two-dimensional techniques), and UV-Vis spectrophotometry. Moreover, solvation studies in dichloromethane and pyridine were performed. The purity of the synthesized porphyrazines was assessed with high-performance liquid chromatography. The UV-Vis studies revealed the presence of characteristic signals for iron(III) porphyrazines as a result of applying an aqueous hydrochloride solution during synthesis. All iron(III) porphyrazines **5a**–**5c** obtained in this work as well as the previously synthesized porphyrazine **5d** were subjected to crystallization experiments in a dichloromethane:methanol (1:1 *v*/*v*) mixture. The X-ray crystal structure analysis of **5d** revealed a saddle-like conformation of the porphyrazine ring. Crystal packing was determined by the octahedral coordination geometry of the central metal ion. All synthesized porphyrazines were also studied using Mössbauer spectroscopy to assess the oxidation and spin state of the iron metal cation inside the macrocyclic core. These studies suggested the presence of the Fe^2+^ cation inside the macrocyclic core axially coordinated by two methanol moieties previously used in solid sample preparation involving slow evaporation of solvent with a high dielectric constant. Moreover, temperature-dependent measurements using this technique showed, for the first time, that the number and flexibility of the peripheral phenyl groups can strongly influence the properties of the iron-binding site and Fe spin state. Their freed rotation at higher temperatures ensures that a high-symmetry binding site of the iron ion in a strong electric field is favored in the Pzs at room temperatures. This is consistent with the results obtained from crystallographic measurements, which showed a single iron site and short Fe-N bond distances (1.926 Å) characteristic of a low-spin Fe^2+^ state [48]. The effects discussed above can be additionally regulated by long-range collective motions of the polymeric structures of Fe-Pzs forming ribbons.

## Data Availability

Not applicable.

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
