# Peer review of "The Valence and Spin State Tuning of Iron(II/III) Porphyrazines with Bulky Pyrrolyl Periphery in Solution and Solid State"

_molecules, 2022, doi:10.3390/molecules27227820_

Round 1

Reviewer 1 Report

The authors report on syntheses, structures, and spectroscopic characterization of several new iron-porphyrazine complexes. Concerning the chemistry involved and the novelty of the whole study, it is of intermediate importance. Therefore I recommend the manuscript to publish in Molecules, but the following concerns should be considered.

In Section 3.2, the iron centers for 5a–c possess +3 oxidation states. However, the iron center of 5a–c used in NMR and Mössbauer measurements are changed from a +3 to a +2 one-electron reduced diamagnetic low-spin species. These one-electron reduced forms (5a’–c’) are clearly different from the isolated forms (i.e., 5a–c). Therefore, the authors should add the synthesis and characterization of the reduced forms in Section 3.2. Otherwise, all presented NMR and Mössbauer spectral data would be suspect.

In connection with the above, the authors should perform cyclic voltammetry measurements of 5a-c to demonstrate that it undergoes a spontaneous one-electron reduction in certain solvents or during recrystallization processes. It is not scientific to mention only the fact that 5a-c is reduced in pyridine and alcohol.

Author Response

Dear Reviewer,

Our response to your comments is in the attached file.

Kind regards,

Tomasz Koczorowski

Reviewer 2 Report

Dear Ms. Yasmine Zhou,

The manuscript (molecules-1978646) entitled ‘’The valence and spin state tuning of iron(II/III) porphyrazines with bulky pyrrolyl periphery in solution and solid state’’ includes synthesis and characterization. This article can be published in the Molecules journal after minor revision.

My suggestions:

Line 184, iron ligands? It is iron-porphyrazines or what is it?

Line 119, The porphyrazine 5c was subjected to oxidation reaction? What is oxidation reagent (did reaction expose to air?)

Line 135, ….substituents [22]. After this sentence, some characteristic proton and 13C-NMR data should be given and interpreted. It should be emphasized that these data are compatible with the literature. I suggest two references at this step: 1- Journal of Porphyrins and Phthalocyanines, 21(03), 231-237, 2017. 2- Journal of Photochemistry and Photobiology A: Chemistry, 405, 112964, 2021.

Line 329, The name of compounds (5a, 5b, and 5c) include chloride ion {for example, [2,7,12,17-Tetrakis(dimethylamino)-3,8,13,18-tetrakis-(2-methyl-5-phenyl-1H-pyrrol-1-yl) porphyrazinato]iron(III) chloride (5a) }. You are adding FeBr2 to the solution not iron chloride. Let's say it's true and it's from HCl, but your drawing in schema 1 has bromide. Please fix them.

Line 344, …….6.57-6.93 (m, 20H, pyrrolyl-H, C6H5), 7.58-7.65 (m, 8H, C6H5) should be written as 6.57-6.93 (m, 20H, C6H5), 7.58-7.65 (m, 8H, pyrrolyl-H, C4H2N). It should be fixed as suggested.

Line 367, …….7.45-7.50 (m, 4H, C6H5) should be written as 7.45-7.50 (m, 4H, pyrrolyl-H, C4HN. Why did you calculate the number of phenyl protons to be 63, it should be 60.

Author Response

(The authors gave the same response as above.)

Reviewer 3 Report

The manuscript “The valence and spin state tuning of iron(II/III) porphyrazines  with bulky pyrrolyl periphery in solution and solid state” after T.Koczorowski, W.Szczolko,  P.Bakun,  B.Wicher, L.Sobotta, M.Gdaniec, A.Teubert, J.Mielcarek, E.Tykarska, J.Korecki, K.Burda, and T.Goslinski reflects a very high level of scientific competence of the authors in electrochemical and catalytic properties of various iron(II/III) porphyrazines and phthalocyanines.

The only one my critical comment is the following. While discussing the  Jahn-Teller effect in 5a compound the authors  should suggest an estimate of potential barrier ΔE. The estimate can be straightforward obtained on basis of the experimental results of the manuscript.

 I recommend the manuscript  for publication in Molecules.

Author Response

(The authors gave the same response as above.)

Reviewer 4 Report

Reviewer comments to submission molecules-1978646 titled “The valence and spin state tuning of iron(II/III) porphyrazines with bulky pyrrolyl periphery in solution and solid state” (authors: Tomasz Koczorowski, Wojciech Szczolko, Pawel Bakun, Barbara Wicher, Lukasz Sobotta, Maria Gdaniec, Anna Teubert, Jadwiga Mielcarek, Ewa Tykarska, Jozef Korecki, Kvetoslava Burda, Tomasz Goslinski)

This article is focused on elucidating the relationship between the bulkiness of perihedral substituents of porhphyrazine moieties and the valence and spin state of iron complexes based on those ligands. The authors of the manuscript are actively and successfully develop the synthesis of various porhphyrazine derivatives for the predictable tuning of their physicochemical properties. Authors managed to trace the change the oxidation state of Fe2+/3+ ion in different solvents by UV-vis spectroscopy and to reveal some features of temperature-dependent distortion of coordination sphere of Fe central atom in solids using the Mössbauer spectroscopy. Since it is hardly possible to show the simultaneous existence of several conformations of the coordination geometry of Fe with different symmetries in solids by some direct method, the arguments of the authors based on experimental data, along with a discussion of references to the corresponding theories, are quite convincing. Their comprehensive and thorough approach to obtaining and analyzing experimental data leads to high quality research with a deep understanding of the nature of the observed phenomena. Although the submitted manuscript mainly describes diamagnetic complexes, the elegant approach developed by the authors on the effect of bulky substituents on the symmetry of the coordination site and intermolecular interactions may be useful in search methods for obtaining new materials that exhibit spin/valence bistability, as well as for SMM and SIM characteristics and other High Spin Molecules features. The presented data should be of interest to specialists studying the design of polyfunctional coordination compounds undergoing a switch in their physical properties. The manuscript seems appropriate for readers of Special Issue "Multifunctional High Spin Molecules and Singlet Biradicals" of Molecules MDPI.

I recommend this article for publication after revision mostly concerning to clarification of some details of the synthesis and the treatment of complexes before the SC XRD and Mössbauer Spectroscopy measurements.

The following comments should be considered:

Most importantly, when reading the text of the manuscript, it is not always clear which complexes and what composition were used for various measurements: [FeIII(Pzs-R)Br] (Scheme 1), [FeIII(Pzs-R)Cl] (Section 3.2. Synthesis, or lines 102), [FeIII(Pzs-R)(MeOH)2] (Section 2.2 X-ray study, lines 147-150), [FeII(Pzs-R)(MeOH)2] (Section 2.3 Mössbauer Spectroscopy, lines 147-150). It is necessary to convey this information to readers more accurately and clearly, to correct all inconsistencies, if any.

·         In particular, I would like to clarify the information on the synthesis:

ü in what form the complexes were obtained after purification by column chromatography, for which formula the analyzes of the composition and structure of the compounds are given, that is for yield, Mp, Rf, UV-vis, NMR, HRMS, HPLC.

ü The authors should explain why the substances under study were not analyzed by elemental analysis and powder XRD.

ü Since complexes in reduced form [FeII(Pzs-R)(MeOH)2] were studied by SC XRD and Mössbauer methods, has their purity and stability been investigated?

·         In connection with the comments made, I also ask you to pay attention to the following passages of the text:

ü  Abstract; Line 18: Did authors use iron(III) derivatives for all appointed measurements???

ü  Abstract; Line 22: It is better to clarify that only 2,5-diphenyl-substituted derivative 5d successfully gave single crystals suitable for SC XRD study.

ü  Section 2.2: lines 147, 150. Fe(III) - ?

I have other comments remarks on the text of the manuscript:

ü  Abstract; Lines 27-29: It is better to point out that for 5d and 5b the contribution of high-spin state fraction was observed. As the sentence is currently written, the reader may be under the delusion that these substances are completely in a high-spin state over the entire temperature range.

ü  Section 2.1 and Section 3.2 When discussing data for UV-Visible, NMR, HRMS, HPLC, references to the relevant figures and tables of SUPPORTING INFORMATION should be added.

ü  Section 2.1 In the discussion of the UV-vis data, a brief comparison with the previously described data on complex 5d should be added.

ü  Section 2.1; Lines 108-122 In the discussion of UV-vis data it is necessary to add some assumptions about which agents are responsible for reduction (Lines 111-112) and oxidation (Line 119) with appropriate references to literature data.

ü  Section 2.3 It is necessary to improve the quality of figures 6-10 in the final version of the manuscript (especially figure 10). In the caption to Figure 10, it is necessary to clarify at what temperature the data obtained by the authors are given. As a wish for future work: it would be interesting to see the temperature dynamics of the components with different symmetry and spin state for expanded set of ligands.

ü  Section 3.1. It is necessary to provide references to articles that describe the synthesis of ligands 4a, 4b, 4c and complex 5d used in this work.

ü  Section 3.2. Since the synthesis of the complexes was carried out according to the same scheme for 5a, 5b, 5c, this part can be combined, leaving only the individual data of the methods of analysis and certification for each compound.

I have several comments on minor changes to the text of the manuscript, mostly concerning confusing expressions and typos.

ü  Abstract; Lines 19-22: the repetition of “subjected”

ü  Abstract; Lines 19-22: “iron substrates” it is better to change to more suitable term

ü  When ions are mentioned, their charge should be given, not their valence (to correct in all text “Fe2+ ion” instead of “iron(II) ion”).

ü  Section 2.1; line 93 and Section 3.2: remove "under diverse conditions" - according to paragraph 3.2, all compounds were purified by column chromatography under the same conditions.

ü  Section 2.3; Line 297: "polymorphism" would be better replaced by a more appropriate term; one polymorph may consist of several statistically distributed (statically or dynamically) different conformations of some molecular or polymeric structural fragments.

ü  The manuscript needs in correction of some other misprints and English grammar inelegancies

Author Response

(The authors gave the same response as above.)

Round 2

Reviewer 1 Report

I reviewed carefully this revised manuscript and the responses by authors to previous reviewers. I believe that the authors have done a good job in addressing comments pointed out by referees. Therefore, the current manuscript would fit very well in Molecules.

Reviewer 4 Report

The text of the manuscript became more suitable for publication after author's amendments. The level of Quality of Representation has increased after the addition of some explanatory comments concerning to clarification of some details of the synthesis and the treatment of complexes before the physicochemical measurements. However, the improved version contains typos and misprints, mainly related to the technical editing of the text (e.g. Line 161: Authors miss correction “Fe(III)” to “Fe(II)” or “Fe2+ cation”).